# An implicit function learning approach for parametric modal regression

**Yangchen Pan, Ehsan Imani**
Univ. of Alberta & Amii
{pan6,imani}@ualberta.ca

**Amir-massoud Farahmand**
Vector Institute & Univ. of Toronto
farahmand@vectorinstitute.ai

**Martha White**
Univ. of Alberta
whitem@ualberta.ca

## Abstract

For multi-valued functions—such as when the conditional distribution on targets given the inputs is multi-modal—standard regression approaches are not always desirable because they provide the conditional mean. Modal regression algorithms address this issue by instead finding the conditional mode(s). Most, however, are nonparametric approaches and so can be difficult to scale. Further, parametric approximators, like neural networks, facilitate learning complex relationships between inputs and targets. In this work, we propose a parametric modal regression algorithm. We use the implicit function theorem to develop an objective, for learning a joint function over inputs and targets. We empirically demonstrate on several synthetic problems that our method (i) can learn multi-valued functions and produce the conditional modes, (ii) scales well to high-dimensional inputs, and (iii) can even be more effective for certain uni-modal problems, particularly for high-frequency functions. We demonstrate that our method is competitive in a real-world modal regression problem and two regular regression datasets.

## 1 Introduction

The goal in regression is to find the relationship between the input (observation) variable $X \in \mathcal{X}$ and the output (response) $Y \in \mathcal{Y}$ variable, given samples of $(X, Y)$. The underlying premise is that there exists an unknown underlying function $g^* : \mathcal{X} \mapsto \mathcal{Y}$ that maps the input space $\mathcal{X}$ to the output space $\mathcal{Y}$. We only observe a noise-contaminated value of that function: sample $(x, y)$ has $y = g^*(x) + \eta$ for some noise $\eta$. If the goal is to minimize expected squared error, it is well known that $\mathbb{E}[Y|x]$ is the optimal predictor (Bishop, 2006). It is common to use Generalized Linear Models (Nelder & Wedderburn, 1972), which attempt to estimate $\mathbb{E}[Y|x]$ for different uni-modal distribution choices for $p(y|x)$, such as Gaussian ($l_2$ regression) and Poisson (Poisson regression). For multi-modal distributions, however, predicting $\mathbb{E}[Y|x]$ may not be desirable, as it may correspond to rarely observed $y$ that simply fall between two modes. Further, this predictor does not provide any useful information about the multiple modes.

Modal regression is designed for this problem, and though not widely studied in the general machine learning community, has been actively studied in statistics. Most of the methods are non-parametric, and assume a single mode Lee (1989); Lee & Kim (1998); Kemp & Silva (2012); Yu & Aristodemou (2012); Yao & Li (2014); Lv et al. (2014); Feng et al. (2017). The basic idea is to adjust target values towards their closest empirical conditional modes, based on a kernel density estimator. These methods rely on the chosen kernel and may have issues scaling to high-dimensional data due to issues in computing similarities in high-dimensional spaces. There is some recent work using quantile regression to estimate conditional modes (Ota et al., 2018), and though promising for a parametric approach, is restricted to linear quantile regression.

A parametric approach for modal regression would enable these estimators to benefit from the advances in learning functions with neural networks. The most straightforward way to do so is to

learn a mixture distribution, such as with conditional mixture models with parameters learned by a neural network (Powell, 1987; Bishop, 1994; Williams, 1996; Husmeier, 1997; Husmeier & Taylor, 1998; Zen & Senior, 2014; Ellefsen et al., 2019). The conditional modes can typically be extracted from such models. Such a strategy, however, might be trying to solve a harder problem than is strictly needed. The actual goal is to simply identify the conditional modes, without accurately representing the full conditional distribution. Training procedures for the conditional distribution can be more complex. Methods like EM can be slow (Vlassis & Krose, 1999) and some approaches have opted to avoid this altogether by discretizing the target and learning a discrete distribution (Weigend & Srivastava, 1995; Feindt, 2004). Further, the mixture requires particular sensitive probabilistic choices to be made such as the number of components.

In this paper, we propose a new parametric modal regression approach, by developing an objective to learn a parameterized function $f(x, y)$ on both input feature and target/output. We use the Implicit Function Theorem (Munkres, 1991), which states that if we know the input-output relation in the form of an implicit function, then a general multi-valued function, under certain gradient conditions, can locally be converted to a single-valued function. We learn a function $f(x, y)$ that approximates such local functions, by enforcing the gradient conditions. We introduce a regularization, that enables the number of discovered modes to be controlled, and also prevents spurious modes. We empirically demonstrate that our method can effectively learn the conditional modes on several synthetic problems, and that it scales well when the input is made high-dimensional. We also show an interesting benefit that the joint representation learned over $x$ and $y$ appears to improve prediction performance even for uni-modal problems, for high frequency functions where the function values change quickly between nearby $x$. To test the approach on real data, we propose a novel strategy to use a regular regression dataset to test modal regression algorithms. We demonstrate the utility of our method on this real world modal regression dataset, as well two regular regression datasets.

## 2  Problem Setting

We consider a standard learning setting where we observe a dataset of $n$ samples, $\mathcal{S} = \{(x_i, y_i)\}_{i=1}^n$. Instead of the standard regression problem, however, we tackle the modal regression problem. The goal in modal regression is to find the set of conditional modes

$$M(x) = \left\{ y : \frac{\partial p(x, y)}{\partial y} = 0, \ \frac{\partial^2 p(x, y)}{\partial y^2} < 0 \right\}$$

where in general $M(x)$ is a multi-valued function. Consider the example in Figure 1. For $x = 0$, the two conditional modes are $y_1 = -1.0$ and $y_2 = 1.0$.

The standard approaches to find conditional modes involve learning $p(y|x)$ or using non-parametric methods to directly estimate the conditional modes. For example, for a conditional Gaussian Mixture Model, a relatively effective approximation of these modes are the means of the conditional Gaussians. More generally, to get precise estimates, non-parametric algorithms are used, like the mean-shift algorithm (Yizong Cheng, 1995). These algorithms attempt to cluster points based on $x$ and $y$, to find these conditional modes. We refer readers to Chen (2018); Chen et al. (2014) for a detailed review.

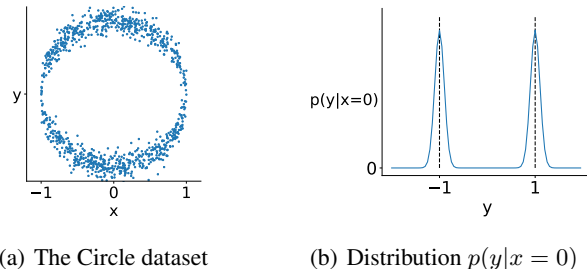

(a) The Circle dataset        (b) Distribution $p(y|x = 0)$

Figure 1: (a) The Circle ataset is a synthetic dataset generated by uniformly sampling $x \in (-1, 1)$, and then sampling $y$ from $0.5\mathcal{N}(\sqrt{1 - x^2}, 0.1^2) + 0.5\mathcal{N}(-\sqrt{1 - x^2}, 0.1^2)$. (b) The bimodal conditional distribution over $y$, for $x = 0$.

## 3  An implicit function learning approach

In this section, we develop an objective to facilitate learning parametric functions for modal regression. The idea is to directly learn a parameterized function $f(x, y)$ instead of a function taking only $x$ as input. The approach allows for a variable number of conditional modes for each $x$. Further, it allows

us to take advantage of general parametric function approximators, like neural networks, to identify these modal manifolds that capture the relationship between the conditional modes and $x$.

## 3.1 Implicit Function Learning Objective

Consider learning an $f(x, y)$ such that $f(x, y) = 0$ for all conditional modes and non-zero otherwise. Such a strategy—finding $f(x, y) = 0$ for all conditional modes—is flexible in that it allows for a different number of conditional modes for each $x$. The difficulty with learning such an $f$, particularly under noisy data, is constraining it to be zero for conditional modes $y_j$ and non-zero otherwise. To obtain meaningful conditional modes $y_1, \ldots, y_{m_x}$ for $x$, the $y$ around each $y_j$ should be described by the same mapping $g_j(x)$. The existence of such $g_j$ is guaranteed by the Implicit Function Theorem (Munkres, 1991)[1] as described below, under one condition on $f$.

**Implicit Function Theorem:** Let $f : \mathbb{R}^d \times \mathbb{R}^k \mapsto \mathbb{R}^k$ be a continuously differentiable function. Fix a point $(a, b) \in \mathbb{R}^d \times \mathbb{R}^k$ such that $f(a, b) = \mathbf{0}$, for $\mathbf{0} \in \mathbb{R}^k$. If the Jacobian matrix $J$, where the element in the $i$th row and $j$th column is $J_{[ij]} = \frac{\partial f(a,b)[i]}{\partial y[j]}$, has nonzero determinant, then there exists open sets $\mathcal{U}_a, \mathcal{V}_b$ containing $a, b$ respectively, s.t. $\forall x \in \mathcal{U}_a$, $\exists$ an unique $y \in \mathcal{V}_b$ satisfying $f(x, y) = \mathbf{0}$. That is, $\exists g : \mathcal{U}_a \mapsto \mathcal{V}_b$. Furthermore, such a function $g(\cdot)$ is continuously differentiable and its derivative can be found by differentiating $f(x, g(x)) = \mathbf{0}$.

We focus on the one dimensional case (i.e. $k = 1$) in this work. The theorem states that if we know the relationship between independent variable $x$ and dependent variable $y$ in the form of implicit function $f(x, y) = 0$, then under certain conditions, we can guarantee the existence of some function defined locally to express $y$ given $x$. For example, a circle on two dimensional plane can be expressed as $\{(x, y) | x^2 + y^2 = 1\}$, but there is no definite expression (single-valued function) for $y$ in terms of $x$. However, given a specific point on the circle $(x_0, y_0)$ ($y_0 \neq 0$), there exists an explicit function defined locally around $(x_0, y_0)$ to express $y$ in terms of $x$. For example, at $x_0 = 0$, we have two local functions $g_1$ and $g_2$: for the positive quadrant we have $y_0 = g_1(x_0) = \sqrt{1 - x_0^2}$ and for the negative quadrant we have $y_0 = g_2(x_0) = -\sqrt{1 - x_0^2}$. Notice that at $y_0 = 0$, the condition required by the implicit function theorem is not satisfied: $\frac{\partial(x^2 + y^2 - 1)}{\partial y} = 2y = 0$ at $y_0 = 0$, and so is not invertible.

Obtaining such smooth local functions $g$ enables us to find these smooth modal manifolds. The conditional modes $g_1, \ldots, g_{m_x}$ satisfy $f(x, g_j(x)) = 0$ and $\frac{\partial f(x, g_j(x))}{\partial y} \neq 0$, where $m_x \in \mathbb{N}$ is the number of conditional modes for $x$. When training $f$, we can attempt to satisfy both conditions to ensure existence of the $g_j$. The gradient condition ensures that for $y$ locally around $f(x, g_j(x))$, we have $f(x, y) \neq 0$. This encourages the other requirement that $f(x, y)$ be non-zero for the $y$ that are not conditional modes at least within the neighborhood of the true mode. We include more discussion about avoiding spurious modes in Section 3.2.

In summary, given a training set $\{(x_i, y_i)\}_{i=1:n}$, we want to push $f(x_i, y_i) = 0$ and $\frac{\partial f(x_i, y_i)}{\partial y} \neq 0$. This naturally leads to the below modal regression objective:

$$L(\theta) \stackrel{\text{def}}{=} \sum_{i=1}^{n} l_\theta(x_i, y_i) \qquad \text{where} \qquad l_\theta(x, y) \stackrel{\text{def}}{=} f_\theta(x, y)^2 + \left( \frac{\partial f_\theta(x, y)}{\partial y} + 1 \right)^2 \qquad (1)$$

We could add hyperparameters for the term constraining the derivative, both in terms of how much we weight it in the objective as well as the constant used to push $\frac{\partial f_\theta(x,y)}{\partial y}$ away from zero. We opt for this simpler choice, with no such hyperparameters. Using a probabilistic perspective, we justify the choice of using square for $f_\theta(x, y)$ and for pushing the partial derivative to $-1$.

**Probabilistic interpretation.** Conventional statistical learning takes the perspective that the training samples are realized random variables drawn from some unknown probability distribution $(X, Y) \sim \mathbb{P} : \mathcal{X} \times \mathcal{Y} \mapsto \mathbb{R}$. For example, we can assume the conditional distribution $p(y|x)$ is Gaussian, resulting in the conventional regression objective where we minimize the mean squared error. Our objective can also been seen as having an underlying probabilistic assumption, but with a weaker assumption.

Consider the following data generation process. We sample $x$ from some unknown marginal, $p(x)$, then sample a mixture component $j$ from $p(j|x)$, and then sample the target $y$ from $p(y|x,j)$. To formally derive our objective, we introduce an oracle: $j : \mathcal{X} \times \mathcal{Y} \mapsto [m_x]$. This function takes a sample $(x, y)$ as input, and returns the corresponding component from which the $y$ value got sampled: $j(x, y)$ is the index of the corresponding Gaussian component. Such a function is not accessible in practice and can only be used for our interpretation here.

We assume that the noise around each conditional mode is Gaussian. More precisely, define $\epsilon(X, Y) \stackrel{\text{def}}{=} g_{j(X,Y)}(X) - Y$. Our goal is to approximate $\epsilon(x, y)$ with parameterized function $f_\theta(x, y)$ for parameters $\theta$. We **assume**

$$\epsilon(X, Y) \sim \mathcal{N}(\mu = 0, \sigma^2) = \frac{1}{\sigma\sqrt{2\pi}} \exp\left(-\frac{\epsilon(X, Y)^2}{2\sigma^2}\right) \tag{2}$$

Note that this is a **weaker assumption** than assuming $p(y|x)$ is Gaussian. For example, $p(y|x)$ can be a mixture of Gaussians. Under our probabilistic assumption, we can estimate $\theta$ by maximum likelihood estimation, giving the objective

$$\arg\min_\theta -\sum_{i=1}^n \ln\left[\frac{1}{\sigma\sqrt{2\pi}} \exp\left(-\frac{f_\theta(x_i, y_i)^2}{2\sigma^2}\right)\right] = \arg\min_\theta \sum_{i=1}^n f_\theta(x_i, y_i)^2.$$

Additionally, we then want to satisfy the constraint on the derivative of $f_\theta$. For an observed $(x, y)$, we have that for some $j \in [m_x]$,

$$\frac{\partial \epsilon(x, y)}{\partial y} = \frac{\partial(g_j(x) - y)}{\partial y} = -1. \tag{3}$$

Therefore, when learning $f$, we encourage $\frac{\partial f_\theta(x,y)}{\partial y} = -1$ for all observed $y$ — as required by the implicit function theorem. In summary, our goal is to minimize the negative log likelihood of $f_\theta$, which approximates a zero-mean Gaussian random variable, under this constraint which we encourage with a quadratic penalty term. This gives the above objective 1.

**Predicting modes.** In modal regression, it is common to either want to find the highest likelihood mode, or to extract all the modes to get a set of predictions. Most existing methods focus only on finding the single most likely mode since they typically have the assumption that the mode is unique (Wang et al., 2017; Feng et al., 2017). The extension to the parametric setting, where we learn a surface characterizing the multi-modal structure, enables us to more obviously handle both cases.

Assume we are given test point $x^*$. We define two sets

$$S_{global}(x^*) = \{y \mid y = \arg\min_y l_\theta(x^*, y)\} \text{ and } S_{local}(x^*) = \left\{y \;\middle|\; \frac{\partial l_\theta(x^*, y)}{\partial y} = 0, \frac{\partial^2 l_\theta(x^*, y)}{\partial^2 y} > 0\right\}$$

For the first case, where we are only interested in the modes with highest likelihood, we need to find the set $S_{global}(x^*)$. For the second case, where the goal is to extract all conditional modes, we need find the set $S_{local}(x^*)$. To understand why these sets are appropriate, notice that we use $l_\theta$ instead of trying simply to find $y$ such that $f_\theta(x^*, y) = 0$. There are two reasons for this: $l_\theta(x^*, y)$ only selects for $y$ that satisfy the implicit function conditions and it also reflects the likelihood of $y$. For the first point, we need to ensure there is an implicit function at the testing point which can map from the input space to the target space.

The second point is more nuanced. Notice that minimizing $l_\theta$ could return all the modes, not just the most likely ones, and so $S_{local}(x)$ and $S_{global}(x)$ should presumably return similar sets. We know that $S_{global}(x^*) \subseteq S_{local}(x^*)$, but ideally we want $S_{global}$ to only consist of the most likely mode. Hence we further constrain the surface with another regularization term, which we introduce in the next section 3.2. In that section, we discuss how, under this regularization, the training points with lower likelihood should have a larger residual, leading to different local loss values. Consequently, $\arg\min_y l_\theta(x^*, y)$ should be unique, or at least should consist of a smaller set of more likely modes.

Finding either of these sets requires a search, or an optimization. In all our experiments, we opted for the simple strategy of searching over 200 evenly spaced values in the range of $y$ to get the two set of predictions. That is, we implement

$$S_{global}(x^*) \approx \{y_j \mid |l_\theta(x^*, y_j) - \min_{i \in [200]} l_\theta(x^*, y_i)| < 10^{-5}, j \in [200]\}$$

$$S_{local}(x^*) \approx \{y_j \mid l_\theta(x^*, y_j) < l_\theta(x^*, y_{j+1}), l_\theta(x^*, y_j) < l_\theta(x^*, y_{j-1}), j \in \{2, ..., 199\}\}.$$

**Connection to energy-based models.** It is worth mentioning that an alternative formulation is possible by combining energy-based models with a margin loss (LeCun et al., 2006). Such model predicts an energy for a pair $(x, y)$. The training process pulls down the energy of pairs of $(x, y)$ in the dataset while the margin constraint avoids the trivial solution of predicting low energy all across the input space. In our method, it is the gradient condition in the implicit function theorem that prevents the trivial solution of predicting a mode everywhere. We may still want to ensure that $l_\theta(x^*, y)$ does not become small where there is no mode, as otherwise our method may predict spurious modes. We address this issue in the next section, with a regularization that we motivate theoretically and empirically.

## 3.2 Higher Order Regularizers to Control the Number of Modes

We first introduce Theorem 1, which motivates our regularization method for avoiding spurious modes. See Appendix A.1 for the proof.

**Theorem 1.** *Let $f(x)$ be a continuously differentiable function s.t. $|f''(x)| \leq u$ for some $u > 0$. Let $a$ be such that $f(a) = \epsilon, f'(a) = k$. Then, $\nexists b$ such that $0 < |b - a| < \frac{2|k|}{u}, f(b) = \epsilon, f'(b) = k$.*

We can consider $a$ in this theorem as a true mode and hence $\epsilon = 0, k = 1$. This theorem is telling us that within $\frac{2}{u}$ to $a$, there is no other point $b$ can make $f_\theta(x^*, b) = 0, \frac{\partial f_\theta(x_i, y_i)}{\partial y} = -1$ if $|\frac{\partial^2 f_\theta(a, y)}{\partial^2 y}| \leq u$ for any $y$ in the target space. The smaller the $u$ is, the farther away the spurious mode has to be. Hence, this theorem tells that the spurious mode should be mostly eliminated by small enough $u$. This suggests a regularization to penalize the second order derivative magnitude:

$$L_\eta(\theta) \stackrel{\text{def}}{=} \sum_{i=1}^{n} f_\theta(x_i, y_i)^2 + \left( \frac{\partial f_\theta(x_i, y_i)}{\partial y} + 1 \right)^2 + \eta \left( \frac{\partial^2 f_\theta(x_i, y_i)}{\partial^2 y} \right)^2 \tag{4}$$

As we increase the weight $\eta$ for the regularizer, the optimization is restricted in the number of modes it can identify. The lowest error choice according to the objective is to find the modes with highest likelihood. Note that this regularizer is not the second order derivative wrt the parameters; it is wrt to the targets. Therefore, it does not introduce significantly more computational cost to compute, and has the same order of computation as the original objective.

The original objective (with $\eta = 0$) actually already has some of the same regularizing affects, due to the regularizer on the gradient w.r.t. $y$. We provide some theoretical support that even with $\eta = 0$, the function is likely to avoid spurious modes in Appendix A.1. However, the original objective does not allow for a mechanism to control this effect, whereas $\eta$ with this second order regularizer provides a tunable knob to more stringently enforce this to be the case. Further, it also allows the user to find a surface that learns only the highest likelihood modes, by picking larger $\eta$. In some cases, even if a mode exists in a multimodal distribution—and so it is not spurious—it may be preferred to ignore it if it has low probability of occurring.

We empirically test the utility of this regularizer for identifying the modes with higher likelihood, and for removing any spurious modes. We generate a training set by uniformly sampling $x \in [-1, 1]$ and training target $y$ by sampling $y \sim 0.8\mathcal{N}(\sqrt{1 - x^2}, 0.1^2) + 0.2\mathcal{N}(-\sqrt{1 - x^2}, 0.1^2)$ if $x < 0$ and $y \sim 0.2\mathcal{N}(\sqrt{1 - x^2}, 0.1^2) + 0.8\mathcal{N}(-\sqrt{1 - x^2}, 0.1^2)$ if $x \geq 0$. That is, the highest likelihood mode is positive if $x < 0$ and is negative if $x > 0$. We generate 4k such points for training and another 1k such points without adding noise to the target for testing.

In Figure 2, we visualize the learned function $l_\theta(0.5, y)$ by training our implicit learning objective with and without regularization respectively. For most points $x^*$, both functions return an accurate set of modes. But, at the more difficult location, in-between when the likelihoods of the modes switches, $x^* = 0.5$, there are more clear differences. At $x^* = 0.5$, the two true conditional modes with high and low likelihood are $\approx -0.87, 0.87$ respectively. Figure 2(b) shows that without the regularization, $\arg\min_y l_\theta(0.5, y)$ possibly returns spurious modes, with larger flat regions; this means we cannot distinguish between the modes with high and low likelihood. In contrast, Figure 2(c) shows that with $\eta = 1$ we identify both the high likelihood mode and the low one by looking at the two local minima at around $\pm 0.87$ of the loss function $l_\theta$ trained with the above regularization. Further, the highest likelihood mode has a smaller $l_\theta(0.5, y)$, and would indeed be the only point in $S_{global}(x^*)$ and correctly identified as the highest likelihood mode. We additionally show the global and local predictions with and without the regularization in Figure 3.

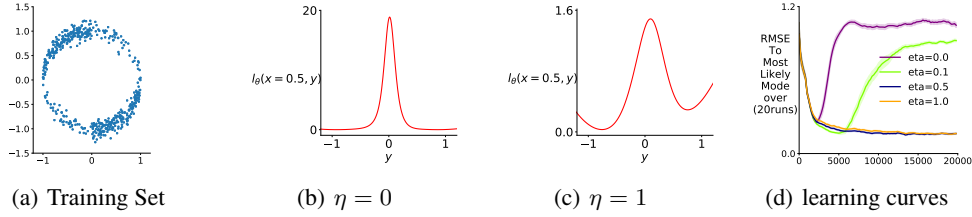

| (a) Training Set | (b) $\eta = 0$ | (c) $\eta = 1$ | (d) learning curves |

Figure 2: (a) shows the training set; (b) shows the loss function $l_\theta(x^* = 0.5, y)$ trained without regularization; (c) shows the one trained with $\eta = 1$. (d) shows the learning curves for $\eta \in \{0.0, 0.1, 0.5, 1.0\}$ in terms of the root mean squared error (RMSE) as a function of number of mini-batch updates on the testing set. The RMSE is computed between randomly picked mode from $S_{global}(\cdot)$ and the corresponding true most likely mode.

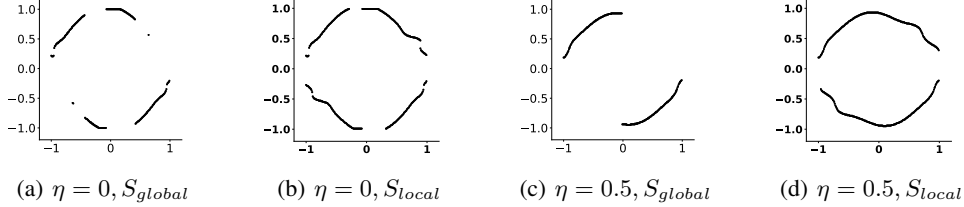

| (a) $\eta = 0, S_{global}$ | (b) $\eta = 0, S_{local}$ | (c) $\eta = 0.5, S_{global}$ | (d) $\eta = 0.5, S_{local}$ |

Figure 3: (a) and (b) shows the predictions of evenly spaced points $x \in \{-1, -0.99, -0.98, ..., 0.99, 1.0\}$ from $S_{global}(\cdot), S_{local}(\cdot)$ learned without using regularization. (c) and (d) shows those points learned with $\eta = 0.5$. (c) shows that the regularizer clearly enables prediction of the most likely mode, whereas (a) has many spurious modes. Looking at both (c) and (d), a decision maker using these predictions could both know the highest likelihood mode (given in (c)) as well as the full set of modes (given in (d)).

Despite some of the issues highlighted in Figure 2 for $\eta = 0$, visualizing the actual predictions made under the original objective reflects a more positive outcome. In Figure 3(a)(b), one can see that the original objective 1 does provide useful predictions even without the regularization; it only fails to identify the highest likelihood mode. The simpler algorithm with $\eta = 0$ looks like a promising choice, if multiple modes are desired, whereas we suggest the use of regularization if the goal is to predict a unique highest likelihood mode. We empirically observe that in many cases, our algorithm is able to achieve competitive performance with other baselines even without using the regularization. We assume $\eta = 0$ for the remainder of this work.

## 4 The properties of implicit function learning

In this section, we empirically investigate the properties of our learning objective. First, on several synthetic Circle datasets, we show the utility of implicit function learning for handling the case where the highest likelihood mode is not unique. Note it is common to assume the existence of a unique mode, to avoid the difficulties of this setting. Second, we demonstrate that our objective allows us to leverage the representational power of neural networks, by testing it on high-frequency data.

### 4.1 Comparison on Simple Synthetic Datasets

**Datasets.** We empirically study the performance of our algorithm on the following datasets. **Single-circle:** The training set contains 4k samples and is generated by the process described in the caption of Figure 1. **Double-circle:** The same number of training points, 4000, are randomly sampled from two circles (i.e. $\{(x, y)|x^2 + y^2 = 1\}, \{(x, y)|x^2 + y^2 = 4\}$) and the targets are contaminated by the same Gaussian noise as in the single-circle dataset. This is a challenging dataset where $p(y|x)$ can be considered as a *piece-wise* mixture of Gaussian: there are four components on $x \in (-1, 1)$ and two components on $x \in (-2, -1) \cup (1, 2)$. The purpose of using this dataset is to examine how the performances of different algorithms change when we increase the number of modes. **High dimensional double-circle.** We further increase the difficulty of learning by projecting the one dimensional feature value to 128 dimensional binary feature through tile coding: $\phi : [-2, 2] \mapsto \{0, 1\}^{128}$. The purpose of using this dataset is to examine how different algorithms can scale to high dimensional inputs.

**Algorithm Evaluation.** We compute RMSE between a randomly picked mode from $S_{global}(\cdot)$ and the closest true mode to it. This evaluation method reflects a practical setting where any of the most likely modes would be acceptable.

**Algorithm Description.** We compare to Kernel Density Estimation (**KDE**) and Mixture Density Networks (**MDN**). KDE is a non-parametric approach to learn a distribution, which has strong theoretical guarantees for representing distributions. It can, however, be quite expensive in terms of both computation and storage. We use KDE because several existing nonparametric modal regression models are proved to converge to the corresponding KDE model (Yao & Li, 2014; Wang et al., 2017; Feng et al., 2017); it therefore acts as an idealized competitor. MDN (Bishop, 1994) learns a conditional Gaussian mixture, with neural networks, by maximizing likelihood. For both methods, we use a fine grid search—200 evenly spaced values in the target space—to find a mode given by the KDE distribution or MDN distribution: $\hat{y} = \arg\max_y \hat{p}(y|x)$. For both our algorithm (**Implicit**) and MDN, we use a $16 \times 16$ tanh units NN and train by stochastic gradient descent with mini-batch size 128. We optimize both algorithms by sweeping the learning rate from $\{0.01, 0.001, 0.0001\}$.

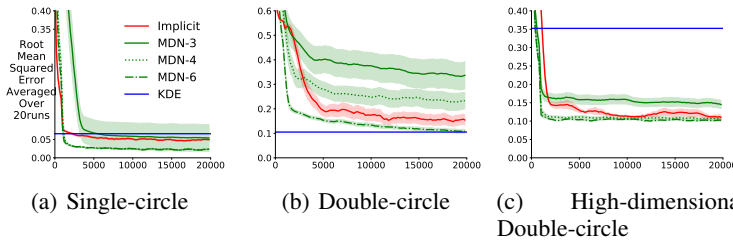

| (a) Single-circle | (b) Double-circle | (c) High-dimensional Double-circle |

Figure 4: (a)(b)(c) show testing RMSE as a function of training steps. MDN-3 indicates MDN trained with 3 components. All results are averaged over 20 random seeds and the shaded area indicates standard error.

The learning curves on the above three datasets are shown in Figure 4. We plot the RMSE as a function of number of mini-batch updates for the two parametric methods MDN and Implicit. KDE is shown as a constant line since it directly uses the whole training set as input. MDN performs poorly with only two components, even when the true number of modes is two; therefore, we include 3, 4 and 6 mixture components. From Figure 4, one can see that: 1) although (a) and (b) have low dimensional feature, MDN with only 3 and 4 components degrades significantly when we increase the number of modes of the training data from two to four. Furthermore, MDN shows significantly larger variance across runs, which we observe frequently across several experiments in our work. 2) KDE scales poorly with both the number of modes and input feature dimension. We also found that it is quite sensitive to the kernel type and bandwidth parameter. 3) Our algorithm Implicit achieves stable performances across all the datasets and performs as well as MDN on the high dimensional double-circle dataset. In contrast, MDN seems to no longer have advantage with a larger number of mixture components than 3 for the high dimensional binary features. We visualize the predictions of our algorithms on the single-circle and double-circle datasets in the Appendix A.3.

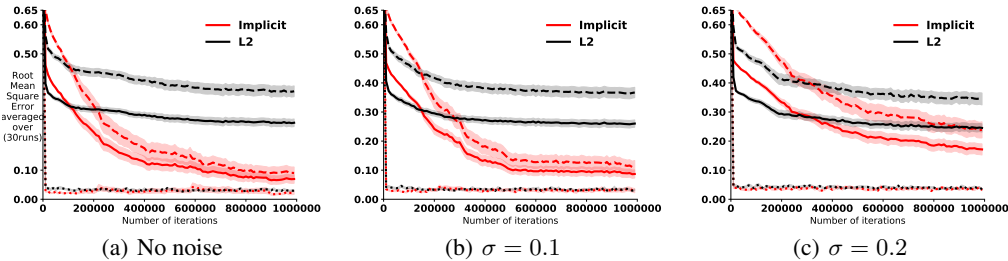

| (a) No noise | (b) $\sigma = 0.1$ | (c) $\sigma = 0.2$ |

Figure 5: Figure (a)(b)(c) show performances of **Implicit (red)** and $l_2$ regression **L2 (black)** objective as we increase the Gaussian noise variance in the High Frequency dataset. We show the testing error measured by RMSE on entire testing set (**solid line**), on high frequency region (i.e. $x \in [-2.5, 0.0)$, **dashed line**) and on low frequency region ($x \in [0.0, 2.5]$, **dotted line**). The results are averaged over 30 random seeds. The targets in the testing set are true targets and hence are not noise-contaminated. We run 1 million iterations to make sure each algorithm is sufficiently trained and both early and late learning behaviour can be examined.

## 4.2 Robustness to high frequency data

The above circle example can be thought of as an extreme case where the underlying true function has extremely high frequency (i.e. when the input changes a little, there is a sharp change for

the true target). In this section, we investigate if our modal regression algorithm does provide an advantage to handle such datasets. We do so by testing it on a uni-modal high-frequency dataset, and comparing the solution found by Implicit to standard $\ell_2$ regression. We generate a synthetic dataset by uniformly sampling $x \in [-2.5, 2.5]$ and compute the targets by $y = \sin(8\pi x) + \xi, x \in [-2.5, 0)$ and $y = \sin(0.5\pi x) + \xi, x \in [0, 2.5]$, where $\xi$ is zero-mean Gaussian noise with variance $\sigma^2$. This function has relatively high frequency when $x \in [-2.5, 0)$ and has a relatively low frequency when $x \in [0, 2.5]$ (see A.3 for more on this dataset). We use a small NN with $16 \times 16$ tanh units. The only difference to the NN for $\ell_2$ is that Implicit has one more input unit, i.e. the target $y$. We perform extensive parameter sweeps to optimize its performance. Please see A.2 for any missing details.

Figure 5(a-c) shows the evaluation curve on testing set for the above two algorithms as the noise variance increases. Notice that, when noise $\xi \equiv 0$, our implicit function learning approach achieves a much lower error (at the order of $10^{-2}$) than the $l_2$ regression does (at the order of $10^{-1}$). As noise increases, the targets become less informative and hence our algorithm's performance decreases to be closer to the $l_2$ regression. Unsurprisingly, for both algorithms, the high frequency area is much more difficult to learn and is a dominant source of the testing error. After sufficient training, our algorithm can finally reduce the error of both the high and low frequency regions to a similar level. We include additional rseults in Appendix A.3, visualizing the learned networks.

# 5 Results on Real World Datasets

In this section, we first conduct a modal regression case study on a real world dataset. Additionally, because our method incorporates information from the target space, we hypothesize it should be beneficial even for regular regression tasks. We therefore also test its utility on two standard regression dataset. Appendix A.2 includes details for reproducing the experiments and dataset information.

## 5.1 Modal Regression: Predicting Insurance Cost

To the best of our knowledge, the modal regression literature rarely uses real world datasets. This is likely because it is difficult to evaluate the predictions from modal regression algorithms. We propose a simple way to construct a datatset from real data that is suitable for testing modal regression algorithms. The idea is to take a dataset with categorical features, that are related to the target, and permute the categorical variables to acquire new target values. Afterwards, we remove those categorical features from the dataset, so that each instance can have multiple possible target values.

We construct a modal regression dataset from the *Medical Cost Personal Dataset* (Lantz, 2013) using the following steps. 1) We determined that the `smoker` categorical variable is significantly relevant to the insurance charge. This can be seen from our Boxplot 6(a). 2) An $\ell_2$ regression model is trained to predict the insurance charges. 3) We flip the `smoker` variable of all examples and query the trained $\ell_2$ model to acquire a new target for each instance in the dataset. 4) We augment the original dataset with these new generated samples. 5) We remove the `smoker` variable from the dataset to form the modal regression dataset.

Each instance has two possible targets, and $\ell_2$ regression can only produce one output; we therefore compute the error between the predicted value and the closest of the two modes. For Implicit and MDN, we compute the error by randomly picking a mode from $S_{global}(\cdot)$ as described in Section 4.1. We report both root mean squared error (RMSE) and mean absolute error (MAE). Figure 6(b)(c) shows the learning curves of different algorithms. It can be seen that: 1) both $\ell_2$ and Huber regression perform significantly worse than our algorithm. This suggests that neither conditional mean nor median is a good predictor on this dataset, and provides evidence that our dataset generation strategy was meaningful. 2) The performance of MDN and KDE are inconsistent across the two error measures. 3) Our algorithm, Implicit, consistently achieves good performance, though it does take longer to learn the relationship.

## 5.2 Standard Regression Tasks

We finally test if our method can achieve comparable performance in standard regression problems. A natural choice for standard regression is to predict the unique mode, with $S_{global}$, as this provides one approach to robust regression. We include the Huber loss as a comparator to see if Implicit has such an effect. We additionally gauge a worst-case scenario, where we predict multiple modes with

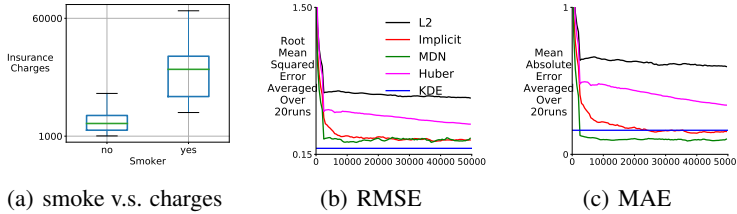

(a) smoke v.s. charges      (b) RMSE      (c) MAE

Figure 6: (a) shows the boxplot of smoking v.s. insurance cost. (b)(c) shows testing error as a function of number of training steps. All results are averaged over 20 random seeds and the standard error is low.

$S_{local}$, but are evaluated based on the furthest distance to the observed target in the dataset (even if in the true underlying data there might be a closer mode).

**Algorithm evaluation.** We report both RMSE and MAE on training and testing set respectively. All of our results are averaged over 5 runs and for each run, the data is randomly split into training and testing sets. For our algorithm, we use $64 \times 64$ tanh units NN. For the $\ell_2$ and Huber regression, we use the same size NN and optimize over unit type ReLu and tanh. Huber regression is used as it is known to be more robust to some skewed or heavy-tailed data distributions.

Baselines and datasets are as follows. **LinearReg:** Ordinary linear regression, where the prediction is linear in term of input features. We use this algorithm as a weak baseline. **LinearPoisson:** The mean of the Poisson is parameterized by a linear function in term of input feature. **NNPoisson:** The mean of the Poisson is parameterized by a neural network (Fallah et al., 2009).

Figure 7 compares the test root mean squared error of different methods. The exact numbers and other error measures are presented in Appendix A.3. We show results on the **Bike sharing dataset** (Fanaee-T & Gama, 2013), where the target is Poisson distributed, in Figure 7(a). The prediction task is to predict counts of rental bikes in a specific hour given 114 features after preprocessing. In Figure 7(b), we show results on the **Song year dataset** (Bertin-Mahieux et al., 2011), where the task is to predict a song's release year by using audio features of the song. The target distribution that is clearly not Gaussian, though it is not clear which generalized linear model is appropriate.

Implicit achieves highly competitive performance on both datasets; although it does slightly worse than MDN on Bike sharing and slightly worse than Huber on Song year dataset when measured by MAE (see A.3). MDN frequently exhibits more variant performance than other algorithms, and its worst case prediction can be significantly worse than from $S_{global}(\cdot)$. In contrast, our algorithm seems to learn an accurate set $S_{local}$, which is only slightly worse that $S_{global}$.

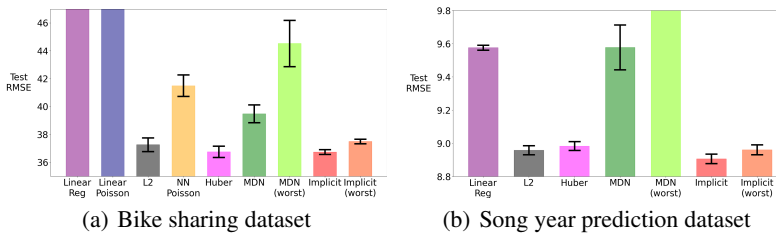

(a) Bike sharing dataset      (b) Song year prediction dataset

Figure 7: Test RMSE on two standard regression tasks. Error bars show standard errors, for 5 runs. For other error measures and more details, see Appendix A.3.

## 6 Conclusion and Discussion

The paper introduces a parametric modal regression approach, using the idea of implicit functions to identify local modes. We show that it can handle datasets where the conditional distribution $p(y|x)$ is multimodal, and is particularly useful when the underlying true mapping has a large bandwidth limit. We also illustrate that our algorithm achieves competitive performance on real world datasets for both modal regression and regular regression. This work highlights the feasibility, and potentially utility, of more widely using modal regression. However, a remaining potential barrier to such widespread use is the difficulty in evaluating modal regression approaches on real data. We provided a strategy to add modality to datasets for evaluation, but did not identify how to use and evaluate mode predictions for standard datasets, even though these datasets likely already have multiple modes. An important next step is to investigate how to evaluate a set of predicted modes, on standard regression datasets.

# 7 Broader Impact Discussion

This work is about parametric methods of modal regression. Non-parametric modal regression methods are typically studied in the statistics community; and there is yet little parametric modal regression algorithm suitable in deep learning setting. Hence, our work should be generally beneficial to the machine learning and statistics research community. Potential impact of this work in real world is likely to be further improvement of applicability of modal regression methods, which provide more information to decision makers than popular regression methods which attempt to learn conditional mean. The proposed objective may be also of high interest to the community studying energy-based models. We do not consider any specific application scenario as the goal of this work.

### Acknowledgements

We would like to thank all anonymous reviewers for their helpful feedback. Particularly, we thank reviewer 2 and the meta-reviewer for carefully reading our paper and providing suggestions to improve the presentation. We also thank Jincheng Mei for reading our theorems and providing suggestions to improve Theorem 1.

### Funding Transparency Statement

We acknowledge the funding from the Canada CIFAR AI Chairs program and Alberta Machine Intelligence Institute.

## Footnotes

[1]The initial version of the theorem was originally proposed by Augustin-Louis Cauchy (1789–1857) and the generalization to functions of any number of real-variables was done by Ulisse Dini (1845–1918).

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
