[Supplementary Material]

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

# A  Appendix

The appendix includes the proof for Theorem 1 and additional theoretical discussions regarding spurious modes in Section A.1. We provide all experimental details for reproducible research in Section A.2 and provide additional empirical results in Section A.3. We also include a brief discussion of some possibly related areas in the end.

## A.1   Proofs and Theoretical Insight

In this section, we provide mathematical proof for our Theorem 1. We then provide additional theoretical intuitions regarding why the spurious modes may not easily show up even without regularization by introducing Theorem 2 and Corollary 1. We conclude this section by some discussions on modal regression algorithm evaluation in Section A.1.3.

### A.1.1   Proof for Theorem 1

**Theorem 1.** Let $f(x)$ be a continuously differentiable function s.t. $|f''(x)| \leq u$ for some $u > 0$. Let $a$ be such that $f(a) = \epsilon, f'(a) = k$. Then, $\nexists b$ such that $0 < |b - a| < \frac{2|k|}{u}, f(b) = \epsilon, f'(b) = k$.

*Proof.* Proof by contradiction. Assume $\exists b \neq a$, such that $|b - a| < \frac{2|k|}{u}, f(b) = \epsilon, f'(b) = k$. By applying first order Taylor expansion to $f(a)$, for some $\xi$, we have

$$f(x) = f(a) + f'(a)(x - a) + \frac{f''(\xi)}{2}(x - a)^2$$

Then,

$$f(b) = f(a) + f'(a)(b - a) + \frac{f''(\xi)}{2}(b - a)^2$$

$$|f'(a)(b - a)| \leq \frac{u(b - a)^2}{2}$$

$$|b - a| \geq \frac{2|k|}{u}$$

This contradicts with $|b - a| < \frac{2|k|}{u}$. This completes the proof. $\qquad\square$

### A.1.2   More discussions regarding the spurious mode

We found that in practice, our method does find correct modes even without the regularization of penalizing the second order derivative, as we discussed in Section 3.2. We introduce below theorem to explain why a spurious mode cannot come out naturally.

**Theorem 2.** *Let $f(x)$ be a continuously differentiable function on the interval $(a, b)$. Let $x_1 < x_2 \in (a, b)$ and $f(x_1) = f(x_2)$ and $f'(x_1)f'(x_2) > 0$ (i.e. the slopes have the same sign and are nonzero). Then $\exists x_0 \in (x_1, x_2)$ such that $f(x_1) = f(x_2) = f(x_0)$ and $f'(x_0)f'(x_1) \leq 0, f'(x_0)f'(x_2) \leq 0$.*

*Proof.* Without loss of generality, let $f'(x_1) < 0, f'(x_2) < 0$. The positive case can be proved in exactly the same way. We prove the existence of such $x_0$ in Part I and prove its derivative property in Part II.

**Part I**. We firstly show $\exists x_0 \in (x_1, x_2)$ such that $f(x_1) = f(x_2) = f(x_0)$. Since $f'(x_2) < 0$, one can choose a $\epsilon_1$ such that $0 < \epsilon_1 < \frac{x_2 - x_1}{2}$ and $f(x_2 - \epsilon_1) > f(x_2)$. Similarly, since $f'(x_1) < 0$, one can choose a $\epsilon_2$ such that $0 < \epsilon_2 < \frac{x_2 - x_1}{2}$ and $f(x_1 + \epsilon_2) < f(x_1) = f(x_2)$. Since the function is a continuously differentiable function, applying intermediate value theorem yields $\exists x_0 \in [x_1 + \epsilon_2, x_2 - \epsilon_1], s.t. f(x_0) = f(x_1) = f(x_2)$.

**Part II**. To show the second part $f'(x_0)f'(x_1) < 0, f'(x_0)f'(x_2) < 0$, consider two cases depending on whether such $x_0$ proved in part I is unique or not.

**Case 1.** When such $x_0$ is unique. Then it implies that there exists no other $x' \in (x_1, x_2), x \neq x_0$ such that $f(x') = f(x_1) = f(x_2) = f(x_0)$. We can prove by contradiction. Assume $f'(x_0)f'(x_1) > 0$.

Then we can replace $x_1$ by $x_0$ in **Part I**'s argument, and this gives another $x_0' \in (x_0, x_2) \subset (x_1, x_2)$ and it contradicts with the prerequisite that $x_0$ is unique. Hence, $f'(x_0)f'(x_1) \leq 0$. Hence, we complete the proof of Case 1 when $x_0$ proved in Part I is unique.

**Case 2.** When such $x_0$ is not unique. Then there are multiple points: $x_1 < x_0' < x_1' < ... < x_2$ such that $f(x_0') = f(x_1') = f(x_1) = f(x_2) = f(x_0)$. Then the first case's result directly applies: one can choose $x_0', x_1'$, i.e. the first closest point and the second closest point to $x_1$; then $f'(x_0')f'(x_1) \leq 0$.

The above two parts complete the proof. □

This theorem immediately leads to the below corollary.

**Corollary 1.** *Let $f(x)$ be a continuously differentiable function on the interval $(a, b)$. If there are, in total $n > 1, n \in \mathbb{Z}$ points: $a < x_1 < ... < x_n < b$ such that $f(x_1) = f(x_2) = ... = f(x_n)$ and they have the same sign of nonzero derivatives, then there must be another $n - 1$ points on $(a, b)$ which have an identical function value with $f(x_i), i = 1, ..., n$ with different derivative signs.*

*Proof.* This is a direct consequence of applying the above Theorem 2 consecutively across $(x_1, x_2)$, $(x_2, x_3)$ etc. □

The corollary tells us that if we have two roots for $f(x) = 0$ on the interval $(a, b)$ and the tangent lines on the two points have the same derivative direction, there must be another root between the two with different derivative sign. Consider the following simple example. We compose a training set by uniformly sampling 4000 data points from a circle: $\{(x, y)|x^2 + y^2 = 1\}$ and train our implicit function $f_\theta(x, y)$. Consider $f_\theta(0, y)$ as the function $f(x)$ in the above Theorem 2. Figure 8 shows the function $f_\theta(x = 0, y), y \in [-1.2, 1.2]$ after training. The correct target values at $x = 0$ should be $+1$ and $-1$. The slope of the function around each of these modes is optimally -1, to satisfy the regularizer $(\frac{\partial f_\theta(x^*, y)}{\partial y} + 1)^2$ which encourages a slope of -1 through all observed $y$. As one can see that, by Theorem 2, there is one root for $f_\theta(0, y) = 0$ at $y = 0$. However, the slope of the function here incurs a high cost for the derivative part.

Now, consider that if there is a wrong prediction $y' \in (-1, 1)$ but $f_\theta(0, y')$ is almost zero and $\frac{\partial f_\theta(0, y)}{\partial y}$ is very close to $-1$—it has the same sign as the other two *correct* predictions. By our Corollary 1, this indicates that there are another two roots for $f_\theta(0, y) = 0$ on $(-1, y'), (y', 1)$ respectively—which leads to two more waves on the interval $(-1, 1)$. That is, every time the objective gives one more incorrect prediction, the function $f_\theta(0, y)$ has two more waves. This would result in a high frequency function, which could be difficult to approximate even with training data. Hence, unless the training data reflects such a high frequency pattern, it is unlikely to choose to do so.

Figure 8: The trained function $f_\theta(x = 0, y)$. The **blue point** is $(0, 0)$; and the other two **black points** are the predicted points by $\arg\min_y f_\theta(0, y)^2 + (\frac{\partial f_\theta(0, y)}{\partial y} + 1)^2$.

### A.1.3 Modal Regression Algorithm Evaluation

To our best knowledge, there is no common standard to evaluate modal regression algorithms when there are multiple true modes. In our main paper, we introduced two ways for evaluation. First, we

randomly pick a predicted mode from $S_{global}(\cdot)$ and compute RMSE/MAE to the closest true mode. Second, on the real world datasets, we consider the worst case prediction from the set $S_{local}(\cdot)$.

We now introduce an arguably more formal, but less intuitive way to evaluate our predictions—Hausdorff distance, which is a natural way to compute distance between two sets. The distance is defined as:

$$d_{\mathrm{H}}(X,Y) = \max\left\{ \sup_{x\in X}\inf_{y\in Y} d(x,y), \sup_{y\in Y}\inf_{x\in X} d(x,y) \right\} \tag{5}$$

where $X, Y$ are two non-empty subsets of some metric space. It should be noted that, when $|Y| = 1$, then this distance is equivalent to computing the worst case prediction as introduced for the regular regression datasets Songyear and Bike sharing in Section 5. This is because $d_{\mathrm{H}}(X,Y)|_{Y=\{y\}} = \max\left\{ \sup_{x\in X}\inf_{y\in Y} d(x,y), \sup_{y\in Y}\inf_{x\in X} d(x,y) \right\} = \max\left\{ \sup_{x\in X} d(x,y), \inf_{x\in X} d(x,y) \right\} = \sup_{x\in X} d(x,y)$. This also indicates that it is only suitable to use this method to evaluate distance between the set $S_{local}(\cdot)$ and the set of true modes when $|S_{local}(\cdot)| > 1$, because when $|S_{local}(\cdot)| = 1$, the evaluation becomes meaningless in that even if it predicts the correct highest likelihood mode, the evaluation will choose to compute its distance to some other true mode as long as the set contains a single prediction. In section A.3, we include learning curves in terms of Hausdorff distance between the set $S_{local}$ and the set of true modes. We do not include the learning curve of Hausdorff distance based on $S_{global}$ as this set frequently contains only a single prediction.

## A.2 Reproduce experiments in the paper

In this section, we provide additional information about datasets we used and experimental details for reproducing all results in this paper. We introduce common settings below.

**Common settings.** Our implementation is based on Python 3.3.6. Our deep learning implementation is based on Tensorflow 1.14.0 (Abadi et al., 2015). All of our algorithms are trained by Adam optimizer (Kingma & Ba, 2015) with mini-batch size 128 and all neural networks are initialized by Xavier (Glorot & Bengio, 2010). For our implicit function learning algorithm, we use tanh units for all nodes in neural network. We search over 200 evenly spaced values for prediction for Implicit, MDN, and KDE. When evaluating the algorithms, the data is randomly split into training and testing sets and we evaluate the testing error every 200 training steps for each random seed unless otherwise specified. The core part of the code is shared at `https://github.com/yannickycpan/parametricmodalregression.git`.

### A.2.1 Reproduce results from Section 3.2

**Algorithm implementation.** We use a relatively larger neural network size than those used on other circle datasets to highlight the effect of our regularization methods. We use $64 \times 64$ tanh units neural network and for each regularization weight $\eta$ value, and optimize learning rate from $\{0.01, 0.001, 0.0001\}$.

**Optimal parameter choice.** The best learning rate is 0.0001.

**Dataset generation.** We generate a training set by uniformly sampling $x \in [-1, 1]$ and training target $y$ by sampling $y \sim 0.8\mathcal{N}(\sqrt{1-x^2}, 0.1^2) + 0.2\mathcal{N}(-\sqrt{1-x^2}, 0.1^2)$ if $x < 0$ and $y \sim 0.2\mathcal{N}(\sqrt{1-x^2}, 0.1^2) + 0.8\mathcal{N}(-\sqrt{1-x^2}, 0.1^2)$ if $x >= 0$. We generate 4k training data points and 1k testing points. For the purpose of plotting prediction in Figure 3, we use the numpy function numpy.arange$(-1, 1, 0.01)$ to generate evenly spaced $x$ values.

### A.2.2 Reproduce results from Section 4.1

**Algorithm implementation.** We keep the neural network size as $16 \times 16$ tanh units and sweep over learning rate from $\{0.01, 0.001, 0.0001\}$ in all circle experiments from Section 4.1. For mixture density network (**MDN**), the maximization is done by using MLE, the method described in the original paper (Bishop, 1994). For **KDE** model, we use the implementation from `https://github.com/statsmodels/statsmodels`. On the high-dimensional double-circle dataset, we consider the binary features as categorical. We input the whole training data to build KDE model. We use normal reference rule of thumb for bandwidth selection. For both KDE and MDN, we use grid search in the target space to find out the mode. That is, $\hat{y} = \arg\max_y \hat{p}(y|x) = \arg\max_y \hat{p}(x,y)$ given testing

| (a) $l_2$ with learning rate $0.01$ | (b) $l_2$ with learning rate $0.001$ |

Figure 9: In (a) we repeat the figure shown in previous Section 4.2.

point $x$. Note that the classic mean-shift algorithm for mode seeking attempts to do hill climbing (i.e. gradient ascent) on a KDE model with multiple initial values, which may still get a local optimum. We opt to directly search from 200 evenly spaced values in the target space to find out the mode given a KDE model.

**Optimal parameter choice.** The best learning rate is $0.01$ for both MDN and Implicit except on double circle dataset, the optimal learning rate is $0.001$ for Implicit on the high dimensional double circle dataset. Note that Implicit performs only slightly worse when learning rate is $0.01$.

**Dataset generation.** Circle dataset is generated by uniformly sampling $x \in [-1, 1]$ first and then $y = \sqrt{1 - x^2}$ or $y = -\sqrt{1 - x^2}$ with equal probability. Double circle dataset is generated by uniformly sampling an angle $\alpha \in [0, 2\pi]$ then use polar expression to compute $x = r \cos \alpha, y = r \sin \alpha$ where $r = 1.0$ or $r = 2.0$ with equal probability. High dimensional double dataset is generated by mapping the original $x$ to $\{0, 1\}^{128}$ dimensional space. We refer to http://www.incompleteideas.net/tiles.html for tile coding software. The setting of tile coding we used to generate feature is: memory size $= 128$, 8 tiles and 4 tilings.

### A.2.3  Reproduce results from Section 4.2

**Algorithm implementation.** We use $16 \times 16$ tanh units neural network for both algorithm. We sweep over $\{0.1, 0.01, 0.001, 0.0001, 0.00001\}$ to optimize stepsize for both the $l_2$ regression and our algorithm, while we additionally sweep over hidden unit and output unit type for the $l_2$ regression from {tanh, relu} and found tanh to be better. The best parameter is chosen to optimize the second half of the learning curve, and the testing error is averaged over 30 random seeds. For the purpose of doing prediction, we use $S_{global}(\cdot)$.

**Optimal parameter choice.** The best learning rate chosen by the $l_2$ regression is $0.01$ while our algorithm chooses $0.001$. As a result, in Figure 9, we also plot the learning curve with learning rate $0.001$ for the $l_2$ regression to make sure that the performance difference is not due to a slower learning rate of our algorithm.

**Dataset generation.** The dataset is generated by uniformly sampling $x \in [-2.5, 2.5]$ and then compute targets according to the equation:

$$y = \begin{cases} \sin(8\pi x) & x \in [-2.5, 0) \\ \sin(0.5\pi x) & x \in [0, 2.5] \end{cases}$$

**Figure generation.** To generate the learning curves from Figure 5 in Section 4.2 and Figure 9, we evaluate the testing error every 10k number of training iterations (i.e. mini-batch updates). We compute the testing errors on the entire testing set, on high frequency area ($x \le 0$) and low frequency area ($x \ge 0$) respectively.

### A.2.4  Reproduce results from Section 5

**Algorithm implementation.** We use $64 \times 64$ tanh units neural network for all algorithms except for NNPoisson where the Poisson mean is using linear activation. We optimize learning rate from

$\{0.01, 0.001, 0.0001, 0.00001\}$. For MDN, we set number of mixture components as 6. When we report the RMSE/MAE on those real world datasets, we choose the average of the best 5 consecutive evaluation testing errors to report. At each run, we evaluate the testing errors every 10k training steps (i.e. mini-batch updates) and we train each algorithm up to 200k steps.

**Optimal parameter choice.** All algorithms choose learning rate 0.0001 except PoissonReg chooses 0.01 on Bike sharing dataset.

**Dataset generation.** The Medical Cost Personal Datasets by Lantz (2013) can be downloaded from `https://github.com/stedy/Machine-Learning-with-R-datasets/blob/master/insurance.csv`. We standardize the *age* and *bmi* variables and perform logarithmic transformation for the target variable *insurance charge* due to its wide range and large values; then we further scale the target variable to $[0, 1]$ when training. We report the errors after converting the predictions back to logarithmic scale. We upload the dataset processed by the steps described in Section 5 through anonymous file sharing and can be downloaded at `https://anonymousfiles.io/WkojSn5F/`, the last two columns are two possible targets/modes.

The bike sharing dataset (Fanaee-T & Gama, 2013) (`https://archive.ics.uci.edu/ml/datasets/bike+sharing+dataset`) and song year dataset (Bertin-Mahieux et al., 2011) (`https://archive.ics.uci.edu/ml/datasets/yearpredictionmsd`) information are presented in figure 11. Note that the two datasets have very different target distributions as shown in Figure 10.

(a) Bike sharing dataset target distribution  (b) Song year dataset target distribution

Figure 10: Bike sharing targets show a clear Poisson distribution while song year dataset's target distribution is not intuitive.

Dataset and preprocessing information

| | Number of instances | Train size | Test size | Input feature dimension after preprocessing | Input feature preprocess | Target preprocess |
|---|---|---|---|---|---|---|
| **Bike sharing** | 17379 | 13903 | 3476 | 114 | remove attributes: date, index, year, weather situation 4 and weekday 7; registered, casual; use one-hot encoding for all categorical variables | Scale to [0, 1] except for poisson regression algorithms; scaled back when compute test error |
| **Song Year** | 515345 | 412276 or 463715 | 103069 or 51630 | 90 | standardize to zero-mean unit variance; statistics acquired by using training set | Scale to [0, 1]; scaled back when compute test error |

Figure 11: Data preprocessing information.

## A.3 Additional experimental results

In this section, we provide the following additional experiments.

1. A.3.1: Visualization of the empirical density function $f_\theta(X, Y)$ after training.
2. A.3.2: Predictions on the single-circle and double circle datasets corresponding to learning curves from Section 4.1.
3. A.3.3: Learning curves in terms of Hausdorff distance for experiments in Section 4.1.
4. A.3.4: Additional experiments to show that our Implicit objective learns a finer neural network representation on the high frequency data as introduced in Section 4.2.
5. A.3.5: Learning curves in terms of Hausdorff distance for experiments on the Modal regression real world dataset in Section 5.
6. The concrete training and testing RMSE and MAE on all regular regression real world datasets.
7. A.3.7: Predictions on a classic inverse problem (Bishop, 1994) in A.3.7.

### A.3.1 Examining the learned error function

It should be noted that our learning objective is based on the assumption that the error $\epsilon(X, Y) \sim \mathcal{N}(\mu = 0, \sigma^2) = \frac{1}{\sigma\sqrt{2\pi}} \exp\left(-\frac{\epsilon(X,Y)^2}{2\sigma^2}\right)$. As a sanity check, we visualize the empirical distribution of the trained $f_\theta(X, Y)$ by showing histograms constructed with all training samples. From Figure 12, one can see that the trained errors do look centering around zero. The one trained with a noise-contaminated dataset show a larger variance (i.e. heavier tail) than the one trained with a noise-free dataset.

(a) Targets in training set are noise-contaminated    (b) Targets in training set are not noise-contaminated

Figure 12: Figure (a) shows the empirical distribution of $f_\theta(X, Y)$ trained with adding zero-mean Gaussian noise with standard deviation $\sigma = 0.1$ and (b) shows the one trained without adding noise to the target.

### A.3.2 Predictions on circle datasets

We further verify the effectiveness of our algorithm by plotting the predictions from $S_{global}(\cdot)$ of evenly spaced 200 $x$ values from $[-1, 1]$ for the single-circle dataset and from $[-2, 2]$ for the double-circle dataset. We do not include predictions from $S_{local}$ as the true modes have equal likelihood. The predictions are shown in Figure 13.

### A.3.3 Hausdorff distance learning curves on circle datasets

Figure 14 shows the result. One can see that: 1) similar to the result from our main paper, MDN is highly sensitive to number of mixture components and seems to have many spurious predictions even on the single circle dataset; 2) KDE is highly sensitive to datatype (i.e. numerical or categorical) and

(a) single-circle        (b) double-circle

Figure 13: (a)(b) shows the predictions of our algorithm on the single-circle dataset by plotting $S_{global}(\cdot)$ for each $x$.

feature dimension; 3) our algorithm Implicit performs stably and reasonably well across different feature dimension and datatype.

(a) Single-circle       (b) Double-circle       (c) high dimension double-circle

Figure 14: (a)(b)(c) show testing Hausdorff RMSE as a function of training steps. MDN-3 indicates MDN trained with 3 components. All results are averaged over 20 random seeds and the shaded area indicates standard error.

### A.3.4    Examining the neural network representation

We further investigate the performance gain of our algorithm by examining the learned neural network representation. We plot the predictions in figure 15(a) and the corresponding NN representation through heatmap in figure 15(b). In a trained NN, we consider the output of the second hidden layer as the learned representation, and investigate its property by computing pairwise distances measured by $l_2$ norm between 161 different evenly spaced points on the domain $x \in [-2.5, 2.5]$. That is, a point $(x, x')$ on the heatmap in figure 15(b) denotes the corresponding distance measured by $l_2$ norm between the NN representations of the two points (hence the heatmap shows symmetric pattern w.r.t. the diagonal). For our algorithm, the target input is given by minimizing our implicit learning objective.

The representations provide some insight into why Implicit outperformed the $l_2$ regression. In figure 15(a), the $l_2$ regression fails to learn one part of the space around the interval $[-2.25, -1.1]$. This corresponds to the black area in the heatmap, implying that the $l_2$ distance between NN representations among those points are almost zero. Additionally, one can see that the heatmap of our approach shows a clearly high resolution on the high frequency area and a low resolution on the low frequency area, which coincides with our intuition for a good representation: in the high frequency region, the target value would change a lot when $x$ changes a little, so we expect those points to have finer representations than those in low frequency region. This experiment shows that given the same NN size, our algorithm can better leverage the representation power of the NN.

### A.3.5    Hausdorff distance learning curves for predicting insurance cost

Figure 16 shows the result. Note that Hausdorff-RMSE means that for each testing point, we use square error for $d(x, y)$ in (5) and then take the root of the mean Hausdorff distance of all testing

(a) Predicted functions        (b) Distance heatmap

Figure 15: (a) Approximated and true functions. (b) The distance matrix showed in heat map computed by hidden layer representation learned by **L2** (left) and **Implicit** (right) method.

points. Similarly, for Hausdorff-MAE, we use absolute difference for $d(x, y)$ then we take the mean Hausdorff distance of all testing points.

(a) RMSE        (b) MAE

Figure 16: Figure (a) (b) shows Hausdorff w.r.t. RMSE and MAE respectively. The results are averaged over 20 random seeds.

### A.3.6 Detailed results on standard regression tasks

Tables 1 and 2 show the performance of the algorithms measured by root mean squared error and mean absolute error.

Table 1: Prediction errors on bike sharing dataset. All numbers are multiplied by $10^2$.

| Algorithms | Train RMSE | Train MAE | Test RMSE | Test MAE |
|---|---|---|---|---|
| LinearReg | 10094.40($\pm$13.60) | 7517.64($\pm$19.95) | 10129.40($\pm$59.26) | 7504.22($\pm$ 44.20) |
| LinearPoisson | 8798.26($\pm$14.58) | 5920.99($\pm$13.66) | 8864.90($\pm$66.07) | 5935.00($\pm$ 38.32) |
| NNPoisson | 1620.46($\pm$47.71) | 1071.39($\pm$29.55) | 4150.03($\pm$77.76) | 2616.49($\pm$ 20.45) |
| L2 | 1919.74($\pm$23.12) | 1421.36($\pm$21.35) | 3726.40($\pm$49.51) | 2526.77($\pm$27.89) |
| Huber | 1914.34($\pm$33.87) | 1398.41($\pm$21.11) | 3675.03($\pm$40.67) | 2487.61($\pm$11.05) |
| MDN | 2888.06($\pm$64.55) | 1456.31($\pm$76.21) | 3948.48($\pm$63.95) | **2298.47**($\pm$36.37) |
| MDN(worst) | 3506.32($\pm$166.30) | 1604.43($\pm$35.90) | 4452.52($\pm$166.65) | 2431.40($\pm$41.31) |
| Implicit | 2075.33($\pm$7.01) | 1504.63($\pm$4.34) | **3674.08**($\pm$16.82) | 2419.54($\pm$12.22) |
| Implicit(worst) | 2154.70($\pm$7.55) | 1602.03($\pm$5.35) | 3749.95($\pm$16.36) | 2514.99($\pm$13.51) |

### A.3.7 A classic inverse problem

One important type of applications of multi-value function prediction is inverse problem. We now show additional results on a classical inverse function domain as used in (Bishop, 1994). The learning dataset is composed as following.

$$x = y + 0.3\sin(2\pi y) + \xi, y \in [0, 1] \tag{6}$$

Table 2: Prediction errors on song year dataset. All numbers are multiplied by $10^2$.

| Algorithms | Train RMSE | Train MAE | Test RMSE | Test MAE |
|---|---|---|---|---|
| LinearReg | 956.40($\pm$0.37) | 681.56($\pm$0.68) | 957.56($\pm$1.49) | 681.66($\pm$1.52) |
| L2 | 850.20($\pm$2.50) | 590.18($\pm$1.63) | 895.77($\pm$2.75) | 608.42($\pm$1.03) |
| Huber | 872.56($\pm$5.00) | 569.49($\pm$1.75) | 898.33($\pm$2.67) | **581.48**($\pm$1.37) |
| MDN | 955.85($\pm$12.97) | 605.58($\pm$4.02) | 957.76($\pm$13.53) | 615.57($\pm$4.37) |
| MDN(worst) | 1699.48($\pm$239.09) | 1212.82($\pm$230.93) | 1700.11($\pm$240.09) | 1215.60($\pm$231.33) |
| Implicit | 876.71($\pm$1.88) | 598.68($\pm$4.11) | **890.61**($\pm$2.87) | 606.91($\pm$3.48) |
| Implicit(worst) | 886.83($\pm$2.92) | 604.16($\pm$2.08) | 896.08($\pm$3.00) | 612.25($\pm$2.53) |

where $\xi$ is a random variable representing noise with uniform distribution $U(-0.1, 0.1)$. We generate 80k training examples. In Figure 17, we plot the training dataset, and predictions by our implicit function learning algorithm with $(\arg\min_y f_\theta(x, y)^2 + (\frac{\partial f_\theta(x,y)}{\partial y} + 1)^2)$. We search over 200 evenly spaced $y$s in $[0, 1]$ for 200 evenly spaced $x \in [0, 1]$ to get points in the form of $(x, y)$s.

(a) Training data    (b) Predictions by Implicit

Figure 17: Figure (a) shows what the training data looks like. (b) shows the predictions of our implicit learning approach.

## A.4 Related Areas

We would like to include a brief discussion about the related areas to our parametric modal regression approach. For the purpose of estimating modes, generative models (Jebara & Pentland, 2002; Li et al., 2020) can be also used. However, learning a generative model may be a more difficult task. It is worth doing a rigorous comparison between our method and some classical generative models in terms of sample efficiency. M-best MAP inference (Batra et al., 2012) may be also adapted to predict conditional modes. A conceptually similar but distinct work is score matching (Hyvärinen, 2005), where the score function can be the derivative of the probability density function w.r.t. data and the score function evaluated at observations in the training set would be pushed to zero.