[Reviews · NeurIPS 2020]

Review 1

Summary and Contributions: The paper proposes a parametric modal regression algorithm to address some shortcomings of “standard” regression approaches that do not deal well with multimodal input. It uses the implicit function theorem to develop an objective, for learning a joint function over inputs and targets and demonstrates (on synthetic data, and a more realistic modal regression problem.) the benefits of the approach.

Strengths: Very nice paper that shows how the Implicit Function Theorem can be used to develop an algorithmic framework. The writing is very clear and provides good background for the problem and existing approaches.

Weaknesses: The experimental results are quite limited; it would be nice to see better comparisons with versions of EM, and also to understand the shortcomings of this approach – when would it fail?

Correctness: Seems correct.

Clarity: Very clear.

Relation to Prior Work: Good, with the exception of the experimental part, where better comparison and a discussion of other methods would be useful.

Reproducibility: No

Additional Feedback: Better experimental part; a more details discussion of when it works and when it doesn't.


Review 2

Summary and Contributions: Update: I've read this paper many times, and I have always had a lot of trouble understanding the mathematical development leading to the objective function. I now understand it better, so I'd like to suggest how I would present it, in case it gives you some ideas for your own presentation: eps(x,y) is the error between y and the "closest mode". Let's define m(x,y) to be a deterministic “mode function” that returns the mode of p(y|x) that is closest to y. So eps(x,y)=m(x,y)-y. We don't know m(x,y), so we can't compute eps(x,y). By modeling assumption, we assert that for fixed x and Y~p(y|x), we have eps(x,Y) ~ N(0,sig^2). Thus, in fact, for a random (X,Y), eps(X,Y)~N(0,sig^2). We want to approximate the function eps(x,y) with a function from the class f_theta(x,y). For a training sample, the likelihood of theta is product_i N(f_theta(x_i,y_i); 0, sig^2). Maximizing this over theta will give an MLE for theta and is equivalent to minimizing sum_i [f_theta(x,y)]^2. Nevertheless, f_theta is unlikely to converge to eps for reasons of identifiability (and potentially other reasons, such as overfitting). To encourage f_theta to approximate eps, we can enforce other conditions on f_theta via regularization. In particular, we know that d eps(x,y)/dy=-1 by Eqn 2, and we can use regularization to encourage the same property in f_theta. Notice I replaced g_j(x) with m(x,y). I don't think g_j(x) works very well. You need g_j(x) to change with y... I think it would be worthwhile to spend a sentence or two giving some characteristics of eps as a function of y for fixed x. Namely, y \mapsto eps(x,y) is a piecewise linear function of y which has slope -1 at all points of continuity, has value 0 at the modes, and has big discontinuities where the mode closest to y changes. At these discontinuities, the absolute value of eps(x,y) will stay the same, but the sign will change. The discontinuities are very important to highlight, because they make it very clear why just taking the zeros of f_theta will lead to spurious modes. Since f_theta is differentiable, it must cross through zero (quickly) at these midpoints between modes. So just looking for zeros isn't sufficient - we need to have a derivative near -1 as well. Similarly, it would be worth stating or illustrating that y \mapsto eps(x,y)^2 looks like a bunch of quadratic U's side by side and all touching 0 at the modes. For the true eps, every local minimum of eps^2 should have value 0. So local minima of eps^2 is another way to find the modes. I think this is a helpful realization for the reader when you discuss taking local minima of regularized f_theta^2 as modes. For some of us reviewers, we thought looking for f_theta = 0 was better-motivated by the paper, although now it's clear why that won't work. Some other points - For equation (2), I think it would be worth noting that this only holds at points of continuity of eps. This will be easy to digest if you also add in the extra explanation of eps and/or an illustration. It would also be nice if there were datasets (artificial is fine) where we can clearly see the gains of this method over the competing methods you consider. ------ This paper proposes a fundamentally new approach to regression problems, in which there are multimodal conditional distributions. The proposed approach is to learn a function f(x,y), which takes both an input and a proposed output, and to produce a value of 0 when the output y is correct for the input (i.e. a conditional mode of y given x), and a nonzero output when y is incorrect. To predict for an input x, one searches over y for the minima of f(x,y) for x fixed to the input. One can either either search for global minima or local minima, depending on the type of mode you are looking for. To prevent the function f from being identically 0, during training we have a penalty term that tries to make the function f(x,y) have a partial derivative w.r.t. y close to -1. They also introduce a penalty to keep the second derivative small, to prevent the function from turning around and returning to zero too quickly when there's no data in a region (which would incorrectly make that area look like a mode). They apply their method to synthetic data as well as real-world datasets.

Strengths: I think the strength of this paper is in the novelty of this approach to multimodal regression. There are fully non-parametric methods, such as kernel density estimation (KDE), which would learn the joint distribution of x and y, and estimate the conditional modes of y given x from that. There are also methods that make strong assumptions about the form of the conditional distribution of y given x, such as mixture density networks, which have a neural network producing the parameters of a mixture of Gaussians for any input x. The method proposed here makes no explicit assumptions about the conditional distribution of y given x, and yet can use any neural network model to estimate f(x,y).

Weaknesses: The theoretical grounding of this work is a bit light. As discussed below, the assumptions being made about the data generating model are not put very clearly, and so the connection with the objective function feels heuristic. Several decisions seem somewhat arbitrary, and alternatives aren't discussed. Some examples are given below. My sense of the empirical results are that, sometimes it's better, and sometimes not, but compared to mixture density networks, the closest competitor, I didn't get a clear sense of when it's better and when not.

Correctness: I don't entirely follow the probabilistic motivation for the objective, as discussed below. But otherwise things seem reasonable.

Clarity: Generally fine, with the exception of the probabilistic section.

Relation to Prior Work: Yes, I think so.

Reproducibility: Yes

Additional Feedback: - Lines 2,3: standard regression with _squared loss_? - "some approaches have opted to avoid this altogether by discretizing the target and learning a discrete distribution": could you use this model as a baseline? Learn a conditional multinomial distribution over the 200 bins of y that you search over? - Line 106 -- You mention negative sampling -- could you mention the pros and cons of this? Your objective function makes no requirements of the function you're fitting outside of the support of the distribution (where you have no training points), and yet we'll sample the values of the function in those out-of-distribution regions (at least out of distibution for y) during prediction. It's worrisome, and negative sampling comes to mind as a remedy. - The definition of epsilon is really unclear. First, the sources of randomness need to clarified. It seems like there's no randomness in epsilon, just in X and Y? Why not define epsilon with lower case x's and y's to clarify that? There's a subscript of j on the RHS but not on the LHS. I guess what you're saying is that each observation Y corresponds to some conditional mode function indexed by j. And we can think of the observation as having been sampled from some Gaussian distribution centered at g_j(X). - At line 112 and 113, you transition from talking about epsilon to talking about f, without explaining the connection. Then you say "Putting this together, our goal is to minimize the negative log likelihood of f_θ, which approximates a zero-mean Gaussian random variable". You say "negative log likelihood of f_θ" -- for this to make sense to me, we must be able to go from f_theta to a likelihood fo the data. But can it be understood in this way? This should be clarified. - In equation (3), it seems arbitrary to add the 2 terms of l together without some weight parameter. Is there any particular reason the penalty on the derivative should be the same scale as the log-likelihood term? - Line 117: Predicting modes. In your development so far, it seems the most obvious way to find the modes would be to minimize f_theta(x,y) rather than l_theta. Is there some motivation for using l? - In the statement of Theorem 1, it's crucial that |f''(x)|<= u _for all x_, which might be worth stating explicitly. The regularization term in (4) only penalizes second derivatives at observed values of y. Yet our search for the modes includes regions of y that may be well outside these support regions. Is this a concern? - The local method for finding the modes S_local is finding the local minima w.r.t. y of l(x,y). Why is this the right thing to do? Given your objective function and problem framing, it seems to take all regions very close to 0. It's not obvious at all why local minima with large values of l are useful points. In Figure 1c, the two local minima correspond to the conditional modes, and your method would find them, but why? Based on your objective function, it seems more like your model is underfitting the modes (because the function doesn't get very close to 0 at .87), and so may be over-regularized? Or do you have some insight over what f converges to and it's not 0 at the modes? - Can you clarify how you are using kernel density estimation? I assume you are using KDE to learn the joint distribution of X and Y, and then using the same techniques to find the maxima for each x (the conditional mode) as you are using to find the minima of your loss function? - I think you should give at least some mention in the main document of how you are measuring performance for hyperparameter tuning. - For the test on the high-frequency data, are you again making predictions by sampling from the 'global modes'? Are the error bands in the figure from that source of sampling randomness and/or from multiple random train/test splits? - Nice design of a pseudo-real dataset. Results aren't so consistent, unfortunately. Seems like Implicit is in some way worse than one of MDN and KDE and never better. It seems like MDN would be hard to beat when you know how many modes there are, and there are relatively few. Is a major advantage of Implicit that you don't have to know how many modes there are? Would be nice if you could find an additional dataset (artificial is fine), where the advantage of Implicit is more clear. - The motivation for using S_local for finding local modes is not at all clear: it's not obvious that there would be connections between local minima of the objective function and local maxima of the conditional density. Is there some evidence that the value of the learned function f(x,y) tracks the condtional densities, at least in the vicinity of the modes? It seems like the most obvious thing to do would be to use S_global for both local and global modes -- if this doesn't work, doesn't it suggest that we're not fitting the data well enough with the objective function? (sorry this is somewhat repetitive of a previous comment) - Suppose your data generating distribution is actually a conditional mixture of Gaussians. Can you figure out, either by theory or experimentation, what your learned function f_theta converges to? Do the modes have value 0 or are they just local minima? I feel like you should at least be able to answer this question, either experimentally or theoretically. This would give a lot more intuition about what's going on here.


Review 3

Summary and Contributions: The paper proposes a method for modal regression, where the goal is to output all the modes of the distribution over outputs conditioned on the input, i.e. p(y|x). Because each input can be associated with many modes, the relation between the input and output is a multivalued function. The main idea is to learn a parametric implicit function over the input and output whose zeros represent the multivalued function. However, there appears to be critical errors in the derivation of the method, and what is actually done is unrelated to this idea.

Strengths: The paper tries to tackle an important problem and presents a nice overview of existing approaches. The problem statement is also clear and precise.

Weaknesses: The derivation of the method contains critical errors (see the "correctness" box for details) and the claimed advantages of the proposed method are somewhat dubious. Below I will explain why a correctly executed version of the method (which is different from the actual method) can fail to realize some of the advantages claimed in the paper. The claimed advantage over approaches that fit a probabilistic model to p(y|x) is stated on L41, which is that such approaches "might be trying to solve a harder problem than is strictly needed". However, conceptually the method is not much different from training an energy-based model (EBM). Specifically, energy in EBMs is also defined on the pair of input and output (x,y), and energy-based models try to assign low energy (high probability) to the observations and high energy (low probability) everywhere else. The only difference with the proposed approach is that EBMs try to make the observation a local maximum of the energy, and the proposed approach tries to make the observation a zero of the learned implicit function f(x,y). However, the proposed method has the same problems as EBMs: (1) it is as hard to find the zeros as the local maxima, and so the modes are hard to find at test time when even the learned energy function or implicit function f(x,y) is given, and (2) negative sampling of (x,y) pairs that are not observed in the training set needs to performed to avoid assigning low energy to non-observations and zero implicit function values to non-observations. The assertion on L106 that ensuring non-zero gradient at observations in the training set obviates negative sampling is incorrect, because this only ensures non-zero implicit function values in the neighborhood. It does not do anything for the y's that are far from the observations, and so could result in spurious modes. The proposed search strategy on L123 of enumerating over different y values to predict the different modes cannot scale to high dimensions, since the number of y values that has to be enumerated grows exponentially in dimensionality.

Correctness: The derivation of the objective in eqn. (3) is incorrect. In particular, the first term in the objective is claimed to be derived from minimizing the negative log-likelihood of the data under the Gaussian noise model in eqn. (1). However, in eqn. (1), the Gaussian is defined in y-space, not the space of implicit function values f(x,y). When derived correctly, the first term should be the squared difference between y and the mode g_j(X). However the mode g_j(X) is not actually known, so it's a chicken-and-egg problem, and there is no straightforward fix. On L109, the noise model \epsilon(X,Y) is not a proper probability distribution unless there is only one mode. For example, if all Y>0 is assigned to a mode g_1(X)=1, and all Y<=0 is assigned to another mode g_2(X)=-1, the density of this noise model would not integrate to 1. On L119, the set of modes S_{local} does not make sense. It is defined to be the set of local minima of the loss function, but the loss function is the summation of the implicit function value squared and the constraint squared. The set of modes should be the set of zeros of the implicit function instead.

Clarity: The exposition of the method is a bit unclear. For example, the implicit function theorem is stated in the general case of d-dimensional output, but the method is presented only in the case of 1D output.

Relation to Prior Work: The reference to implicit function theorem is incorrect - it is a classical result in analysis and doesn't need to include a reference. If a reference is desired, credit should go to Augustin-Louis Cauchy and Ulisse Dini, and the textbook reference should accompany the statement of the theorem on L88-92. L36-47 only covers work on mixture models, but the large body of work on probabilistic/generative models can be used for modal regression. They are parametric models and so could be considered as related work. For example, energy-based models like CRFs can be used to model p(y|x), and M-best MAP inference [a] could be done to output the different modes. Alternatively, deep generative models can be used to model p(y|x) and a learning algorithm specifically designed to cover all modes can be used to make the model produce different modes when sampled [b]. At a conceptual level, the proposed method is related to score matching [c], since its goal is to make the score function (derivative of the probability density w.r.t the data) evaluated at observations in the training set close to zero and non-zero elsewhere. The score can be viewed as an analogue of the implicit function proposed by the paper, but with a probabilistic interpretation. [a] Batra et al., "Diverse M-Best Solutions in Markov Random Fields", ECCV 2012 [b] Li et al., "Multimodal Image Synthesis with Conditional Implicit Maximum Likelihood Estimation", IJCV 2020 [c] Hyvärinen et al., "Estimation of non-normalized statistical models by score matching", JMLR 2005

Reproducibility: Yes

Additional Feedback:


Review 4

Summary and Contributions: This submission proposes an approach to learn an implicit function for parametric model regression in the neural network framework.

Strengths: The targeted research problem is interesting and overall the proposed approach looks reasonable. This work probably attracts some participants in the NeurIPS community. How to understand the sentence on page

Weaknesses: The paper is not so well written and not easy to be followed by readers with limited related background. Also, to understand the proposed approach, the authors may want to clearly explain each equation. For instance, what is the p function on page 2 line 65? How to understand the sentence on page 3 line 99? Should the y be changed to y_i in Eq. (4) on page 4 line 140? Why there is no weight for the second term? Why can we implement S_{global} using the equation on page 4 line 124? It looks like the implementation may not be very efficient for a large scale dataset. And this gives within-sample solution, which may not be accurate for unevenly distributed data. In Figure 1, the learning curves go up eventually for eta = 0.0 and 0.1. Does this indicate we need to carefully experiment the weights for different datasets when using this method?

Correctness: Seems to be correct.

Clarity: Not easy to follow.

Relation to Prior Work: Yes.

Reproducibility: Yes

Additional Feedback:

[Author Response · NeurIPS 2020]

We thank all reviewers for their helpful and constructive feedback. We will address all minor issues as everyone
mentioned. From both positive and negative reviews, we believe all reviewers read our paper carefully, and we
appreciate it. Should we address your main concerns well, we hope you improve your score accordingly.

**To all, particularly to R3 who misunderstands our error assumption and derivation of the objective**. In statistical
learning, a training example $(x, y)$ is a realized random variable $(X, Y)$ drawn from some unknown probability
distribution $\mathbb{P}(x, y)$. In regression, we do not need to model $\mathbb{P}$, but rather only $p(y|x)$. For example, in $l_2$ regression,
we assume $p(y|x)$ is Gaussian: $p(y|x) = \mathcal{N}(y; g(x), \sigma)$, where $g(x)$ is the mean and $\sigma$ the standard deviation (STD).
Under this assumption, the error $g(X) - Y$ is Gaussian, irrespective of what the marginal distribution of $X$ is. In
contrast, our key idea is to **avoid** making assumptions on $p(y|x)$. Instead, we only assume **the error** $g(X) - Y$ is
Gaussian. Such an assumption is weaker: if $p(y|x)$ is Gaussian, then the error must be Gaussian; however, if the error is
Gaussian, $p(y|x)$ does not have to be Gaussian. For example, for Gaussian mixture $p(y|x) = \sum_{i=1}^{k} w_i N(y; g_i(x), \sigma)$,
the error $\epsilon(X, Y) = g_i(X) - Y$ is Gaussian with zero mean and variance $\sigma^2$: $p(\epsilon(x, y)) = N(\epsilon(x, y); 0, \sigma)$.

Then we propose to use a function $f_\theta(X, Y)$ to approximate the error $\epsilon(X, Y)$. This can be accomplished by maximizing
the likelihood that $f_\theta(X, Y)$ is a zero-mean Gaussian for the given data. This objective has a trivial solution. Hence, the
implicit function theorem is applied to ensure there exists some implicit function to express $y$ in terms of $x$ around each
training point, and results in the additional term $(\frac{\partial f_\theta}{\partial y} + 1)^2$. As R2 pointed out, we could add a parameter to weight the
two parts differently. For this work, we opted for the simplest approach with fewer hyperparameters.

**To R2, R3, R4**. Let us clarify how to make predictions and $S_{local}$. The loss $l_\theta$ is used, instead of finding $f_\theta(x, y) = 0$,
because it encodes the full constraints during learning that were used to identify the modes. Finding points $y$ that have
low $l_\theta(x, \cdot)$ ensures we find the most plausible set of conditional modes. As mentioned in the text, arguably one of the
most important limitations of this approach is that it might find spurious modes. We address this issue explicitly, in
Section 3.2, both proposing a method to reduce the likelihood of spurious modes and showing that the simpler approach
is often itself quite robust to spurious modes. The strategy to avoid such spurious modes relies on using $l_\theta$ for prediction,
to sufficiently constrain the set of possible candidates. As for $S_{local}$, our idea is to take advantage of the residual. It is
different with conventional $l_2$ regression, where the prediction function is fixed after training. Our training process
can be thought of as constructing many implicit prediction functions. Those prediction functions are defined by both
parameters $\theta$ and the input $(x, y)$ itself. When searching for $y$ given an input $x$, it is unlikely to reconstruct an prediction
function which is exactly the same as one of those learned during training. That's why $S_{local}$ makes sense: those modes
with higher likelihood should have lower residual.

**To R2, R3**. L106 negative sampling. We actually meant negative sampling is problematic and we avoid it.

**To R2.** For some of your remaining concerns. 1) Thm 1, out of support issue. In theory, Thm1 tells that the spurious
mode should be mostly eliminated by small enough $u$. In practice, the empirical results show that the gradient condition
by itself often fixes the problem of spurious modes. 2) We use KDE to learn the joint distribution and use the same
way to find maxima for each $x$. 3) Hyperparameter tuning. There is no standard way to do model selection in modal
regression. Hence the best thing we can do is to ensure fair comparison: we optimize each algorithm's testing error
by choosing best parameter setting. In fact, we sweep over larger range of hyperparameters for our competitors. We
can definitely include some discussions. 4) High-frequency dataset. Yes, we use global mode. Source of randomness:
both. 5) Where $f_\theta$ converges to. At the modes, $f_\theta$ is almost zero. The objective pushes $f_\theta$ outside of the modes to be
non-zero, and encourages $f_\theta$ to have a derivative of 1 as much as possible between modes. We have figures showing $f_\theta$
and can definitely add them.

**To R3**. 1) EBMs are clearly different. They model the joint distribution of $(X, Y)$, but we model the error. You
also claim that our method has the same problem as EBMs:"negative sampling of $(x, y)$ pairs ... " However, we
**never** do negative sampling, it only appears one time in the paper. Our approach actually avoid negative sampling. 2)
Other probabilistic models. Like other regression approaches, we are motivated by the principle: model only what
you need, and not more. MDN is a reasonably representative probabilistic model—which is why its chosen for the
experiments—but we can highlight a few more models in the intro that could be used for this problem. 3) We will cite
the given references, including about score matching which is different but relevant.

**To R3, R4**. There is some concern about the efficiency of the approach. In this first work, our primary goal was to
investigate the viability of this (first) parametric approach to modal regression. We have not yet focused on smarter
algorithms for obtaining $y$ during prediction. The method intuitively scales well with increased dimensionality in $x$ and
dataset size, in contrast to previous nonparametric modal regression algorithms. This already is a victory, and facilitates
the use of modal regression in a broader range of settings. For higher-dimensional outputs, we do at least have an
obvious strategy of gradient descent to search for minimal $y$; but more work needs to be done to understand scalability
for higher-dimensional outputs.

[Meta-Review · NeurIPS 2020]

Thank you for your submission to NeurIPS. This was a borderline paper, with three of the reviewers being position (including one that went from negative to positive after the discussion period), and one remaining quite unconvinced about the paper. Having read through the paper myself, I agree with the overall assessment that there are both some strong and weak aspects to the paper. On the positive side, I found the proposed modeling approach here genuinely interesting. The idea of parameterizing a kind of energy function (I know the authors don't use this notion, but I do feel it is warranted ... this refers to the loss function that is ultimately minimized to find the modes) as the combination of the squared value of a the function f plus the penalty of the function derivative away from zero, this was rather interesting, and I believe likely to have applications well beyond just modal regression. At this same time, I agree with some of the reviewers that the presentation and derivation of this function seems rather strange: the authors motivate finding points where f(x,y) = 0, but of course these aren't the actual points of interests to find, since the "true" function f will be discontinuous, and approximations will have a sharp positive slope away from data so that the function can then have a zero with negative slope at all the modes. It seems much more natural to me to simply parameterize the proposed loss function as a kind of energy function that obviates the need for negative sampling by means of the gradient penalty. Obviously, the authors should present this as they feel best suits the paper, but this point in particular caused a great deal of discussion amongst the reviewers, and was one of the chief reasons for the disagreement. I would strongly encourage the authors to consider the alternative rationale about why minimizing the full regularized loss term rather than f alone is a more appropriate target, especially since the entire method is originally motivated by finding zeros of f, yet ultimately this is not what is done. The other weakness of the paper is frankly that the presented results are underwhelming (it's unclear that it's performing any better than alternative approaches on the real data examples, and even these are quite limited and straightforward), though I believe that the presented approach may still be valuable even with rather preliminary results. If there is any way to conduct additional experiments to validate the proposed method, especially on real problems with multi-modal response variables, this would greatly strengthen the paper. Ultimately, I believe that these strengths make the paper worth acceptance, but I strongly encourage the authors to address all the points above.